# Common Variants in *IL-20* Gene are Associated with Subclinical Atherosclerosis, Cardiovascular Risk Factors and IL-20 Levels in the Cohort of the Genetics of Atherosclerotic Disease (GEA) Mexican Study

**DOI:** 10.3390/biom10010075

**Published:** 2020-01-03

**Authors:** Javier Angeles-Martínez, Rosalinda Posadas-Sánchez, Eyerahi Bravo-Flores, María del Carmen González-Salazar, Gilberto Vargas-Alarcón

**Affiliations:** 1Department of Molecular Biology, Instituto Nacional de Cardiología Ignacio Chávez, Mexico City 14080, Mexico; jabojangeles@yahoo.com.mx; 2Department of Endocrinology, Instituto Nacional de Cardiología Ignacio Chávez, Mexico City 14080, Mexico; rossy_posadas_s@yahoo.it (R.P.-S.); telesforo_13@yahoo.com.mx (M.d.C.G.-S.); 3Department of Immunobiochemistry, Instituto Nacional de Perinatología Isidro Espinosa de los Reyes, Mexico City 11000, Mexico; eyerahiqfb@yahoo.com.mx

**Keywords:** cardiovascular risk factors, genetic association, inflammation, interleukin 20, polymorphisms, subclinical atherosclerosis

## Abstract

Inflammation has been involved in the development of atherosclerosis, type 2 diabetes mellitus, insulin resistance, and obesity. Interleukin 20 is a pro-inflammatory cytokine encoded by a polymorphic gene located in chromosome 1. The aim of the study was to evaluate the association of two *IL-20* polymorphisms (rs1400986 and rs1518108) with subclinical atherosclerosis (SA), cardiovascular risk factors and IL-20 levels in a cohort of Mexican individuals. The polymorphisms were determined in 274 individuals with SA and 672 controls. Under different models, rs1400986 (OR = 0.51, *P*_codominant1_ = 0.0001; OR = 0.36, *P*_codominant2_ = 0.014; OR = 0.49, *P*_dominant_ = 0.0001 and OR = 0.55, *P*_additive_ = 0.0001) and rs1518108 (OR = 0.62, *P*_codominant2_ = 0.048 and OR = 0.79, *P*_additive_ = 0.048) were associated with a lower risk of SA. These polymorphisms were associated with cardiovascular risk factors in individuals with SA and controls. Controls with the rs1400986 *TT* genotype presented high levels of IL-20 (*p* = 0.031). In individuals with the rs1400986 *CC* genotype, we observed a negative correlation between IL-20 levels and total abdominal tissue (TAT), visceral abdominal tissue (VAT) and subcutaneous abdominal tissue (SAT). Our results indicate that the *IL-20* rs1400986 and rs1518108 polymorphisms were associated with decreased risk of developing SA and with some cardiovascular risk factors in individuals with SA and healthy controls. Negative correlation between BMI and VAT/SAT ratio in individuals with rs1400986 *CC* genotype and among IL-20 levels and TAT, VAT and SAT was observed.

## 1. Introduction

Cardiovascular disease, principally coronary artery disease (CAD) is the leading cause of preventable death worldwide [1]. A major reason for this trend is the ongoing epidemic of type 2 diabetes mellitus (T2DM) and obesity-induced insulin resistance (IR) [2]. Substantial evidence shows that IR is associated with CAD risk factors and is likely a common ground for the diabetic atherogenic milieu [3]. Further, IR and T2DM are thought to be mechanistically linked to CAD via subclinical atherosclerosis (SA) [4]. SA develops over several decades and often remains asymptomatic until the occurrence of an acute, life-threatening event. Two subclinical measures of atherosclerosis have been used to predict CAD. One is carotid intima-media thickness (cIMT), a measure of the intimal and medial layers of the carotid artery walls and the other is coronary artery calcification (CAC), a marker of subclinical coronary atherosclerosis [5,6]. Visceral adipose tissue (VAT) accumulation is clearly associated with a higher risk of T2DM and CAD and is positively associated with cardiovascular risk factors [7,8,9]. It is well known that inflammation plays an important role in the development of T2DM, IR, obesity and CAD [10,11,12]. IL-20 is a pro-inflammatory cytokine produced preferentially by monocytes [13]. It belongs to the IL-10 family, which also includes IL-19, IL-22, IL-24, IL-26, IL-28 and IL-29 [14]. This cytokine acts stimulating the angiogenic activity through the induction of vascular endothelial growth factor (VEGF) and IL-8 production. Chen et al. [15] analyzed the expression of IL-20 and its receptor complex in human and mice (apolipoprotein E-deficient) atherosclerotic lesions. The results of this study suggest that IL-20 is a proatherogenic cytokine that contribute to the progression of the disease. The *IL-20* gene is located in chromosome 1 and two polymorphisms (rs1400986 and rs1518108) have been associated with inflammatory diseases, such as, psoriasis and ulcerative colitis [14,15]. Also, these polymorphisms were associated with chronic hepatitis B infection in African-Americans [16]. On the other hand, an “in silico” analysis that we made showed that the rs1400986 polymorphism modify a binding site for the MZF1 transcriptional factor, having a possible functional effect. Despite the important role of this cytokine in the inflammatory process and in consequence in the development of atherosclerosis [15,17], at the present, there are not studies that analyzed the association of the polymorphisms located in the gene that encodes this cytokine with the presence of atherosclerosis and cardiovascular risk factors. Thus, the aim of the present study was to evaluate the association of rs1400986 and rs1518108 *IL-20* polymorphisms with SA and cardiovascular risk factors in a Mexican population.

## 2. Materials and Methods

### 2.1. Study Population

The study complies with the Declaration of Helsinki and was approved by the Ethics Committee of the Instituto Nacional de Cardiología Ignacio Chávez (INCICH). All participants provided written informed consent. Study participants were a subset of the Genetics of Atherosclerotic Disease (GEA) Mexican Study (*n* = 946) population. To be included in the study, volunteers were apparently healthy and asymptomatic without family history of premature CAD. Participants were recruited from blood bank donors and through brochures posted in social services centers. Computed tomography (CT) scans of the chest and abdomen were performed using a 64-channel multidetector helical computed tomography system (Somatom Cardiac Sensation, 64, Forcheim, Germany) and interpreted by experienced radiologists. Scans were read to assess and quantify various parameters: (a) total abdominal tissue (TAT), subcutaneous abdominal tissue (SAT) and visceral abdominal tissue (VAT) areas as previously reported by Kvist et al. (1988) [18]; (b) liver to spleen attenuation ratio (L:SAR) as described by Longo et al. (1993) [19]; and (c) coronary arterial calcification (CAC) score using the Agatston method [20]. Fatty liver was defined as L:SAR ≤ 1.0. In all individuals, clinical, demographic, anthropometric and biochemical parameters were evaluated as previously described [21,22,23]. Exclusion criteria were congestive heart failure, liver, renal, thyroid or oncological disease and premature CAD.

### 2.2. Definition of Subclinical Atherosclerosis

CAC quantified by the Agatston score has been known to be an excellent biomarker of atherosclerosis, independently predicting clinical outcomes, such as coronary heart disease [24,25,26]. In our group, after performing the computed tomography scans, 274 individuals were classified into the SA group (individuals with a CAC score > 0), while 672 participants comprised the healthy control group (CAC score = 0).

### 2.3. Quantification of IL-20 Concentration

In a subsample of 106 control individuals, IL-20 plasma concentrations were quantified. For the determination of the IL-20 levels, we designed a panel, which also included the IL-19 and IL-22 cytokines (Bio-Rad, Hercules, CA, USA). The levels were detected using Luminex multi-analyte technology (Bio-Plex ProTM, Bio-Rad, Hercules, CA, USA) according to the manufacturer’s instruction. Before starting the bioassay, the samples were thawed on ice and once ready for use, they were centrifuged at 10,000 rpm for 4 min. Samples were incubated with antibodies immobilized on color-coded microparticles, washed to remove unbound material and then incubated with biotinylated antibodies to the molecules of interest. After further washing, the streptavidin-phycoerythrin conjugate that binds to the biotinylated antibodies was added before the final washing step. The Luminex analyzer was used to determine the magnitude of the phycoerythrin-derived signal in a microparticle-specific manner. The data were analyzed using Bio-Plex Manager software. Data were expressed in pg/mL.

### 2.4. Genetic Analysis

Genomic DNA from whole blood containing ethylenediamine-tetra-acetic acid was isolated with a no enzymatic method [27]. According to the manufacturer’s instructions (Applied Biosystems, Foster City, CA, United States), the rs1400986 (C___1747382_10) and rs1518108 (C___1747381_10) *IL-20* polymorphisms were determined in genomic DNA using 5′ exonuclease TaqMan genotyping assays on an ABI Prism 7900HT Fast Real-Time polymerase chain reaction (PCR) system. Ten percent of the samples were determined twice in order to corroborate the assignment of the genotypes.

### 2.5. Statistical Analysis

Data are expressed as the mean (standard deviation), median (interquartile range) or frequencies. Normality of distribution was tested by the Kolmogorov-Smirnov test. The differences in continuous variables between groups were assessed with t-Student’s test and Mann-Whitney U test, as appropriate. Nominal variables were analyzed using the chi-squared test. The analysis was made using the SPSS version 20.0 statistical package (SPSS, Chicago, II, USA). The associations of the polymorphisms with SA and cardiovascular risk factors were analyzed using logistic regression under the following inheritance models: additive (add-major allele homozygotes vs. heterozygotes vs. minor allele homozygotes), codominant 1 (cod1-major allele homozygotes vs. heterozygotes), codominant 2 (cod2-major allele homozygotes vs. minor allele homozygotes), dominant (dom-major allele homozygotes vs. heterozygotes+minor allele homozygotes) and recessive (rec-major allele homozygotes+heterozygotes vs. minor allele homozygotes). When the association with SA was tested, the models were adjusted for age, gender, body mass index, current smoking status, alanine aminotransferase, aspartate aminotransferase and uric acid. To evaluate the association with cardiovascular risk factors, the models were adjusted for age, gender and BMI. Bonferroni correction was used for multiple testing due to the increased risk of Type I error (0.05 divided by 5; *P* < 0.01). Statistical power to detect associations of polymorphisms with SA exceeded 0.80 as estimated with QUANTO software (http://hydra.usc.edu/GxE/). Genotype frequencies did not deviate from Hardy-Weinberg equilibrium in any case (HWE, *p* > 0.05). Pairwise linkage disequilibrium (LD, D’) estimations between polymorphisms and haplotype reconstruction were performed with Haploview version 4.1 (https://www.broadinstitute.org/haploview/haploview) (Broad Institute of Massachusetts Institute of Technology and Harvard University, Cambridge, MA, USA).

## 3. Results

### 3.1. Study Samples Characteristics

Table 1 summarizes the baseline characteristics of the study groups. After all the experiments and the data compilation, 274 individuals with SA (199 males and 75 females) and 672 healthy controls (257 males and 415 females) were included in the final analysis. Descriptive statistics of the study participants and differences in median by clinical features for each predictor variable are provided in Table 1 and Table 2. In comparison with the healthy control group, individuals with SA exhibited higher levels of systolic and diastolic blood pressure, VAT, SAT, total-cholesterol (TC), low density lipoprotein-cholesterol (LDL-C), triglycerides, non-high density lipoprotein-cholesterol (non-HDL-C), gamma-glutamyl transpeptidase (GGT), apolipoprotein B, glucose, homeostasis model assessment of insulin resistance (HOMA-IR), creatinine, adiponectin, uric acid and albumin (Table 1).

Additionally, in comparison with the control group, participants in the SA group showed a higher prevalence of hypercholesterolemia, LDL-C > 130 mg/dL, hypertriglyceridemia, T2DM, IR, metabolic syndrome, hypertension, high VAT, fatty liver and hyperuricemia (Table 2).

### 3.2. Association of Polymorphisms with SA

Genotype distribution of the polymorphisms in individuals with SA and controls were in accordance with the HWE expectation (*p* > 0.05). Genotype distribution in the study groups is presented in Table 3. Under several inheritance models, adjusted by age, gender, body mass index, current smoking status, alanine aminotransferase, aspartate aminotransferase and uric acid, the rs1400986 polymorphism was significantly associated with a low risk of developing SA (OR = 0.51, 95% CI: 0.36–0.73, *P*_cod1_ = 0.0001; OR = 0.36, 95% CI: 0.16–0.81, *P*_cod2_ = 0.014; OR = 0.49, 95% CI: 0.35–69, *P*_dom_ = 0.0001 and OR = 0.55, 95% CI: 0.41–0.73, *P*_add_ = 0.0001). In addition, the rs1518108 polymorphism showed a marginally significant association with a low risk of developing SA (OR = 0.62, 95% CI: 0.39–99, *P*_cod2_ = 0.048 and OR = 0.79, 95% CI: 0.63–99, *P*_add_ = 0.048). Thus, both polymorphisms were independently associated with a lower risk of developing SA.

### 3.3. Association of the IL-20 Polymorphisms with Cardiovascular Risk Factors

Table 4 shows the association of *IL-20* polymorphisms with some cardiovascular risk factors in (i) controls and (ii) SA individuals. In healthy controls, rs1400986 was significantly associated with high levels of inflammation (hsCRP ≥ 3mg/L, OR = 1.45, 95% CI: 1.01–2.10, *P*_cod1_ = 0.047) and with low levels of GGT > p75 (OR = 0.41, 95% CI: 0.19–0.88, *P*_cod2_ = 0.023 and OR = 0.42, 95% CI: 0.19–0.89, *P*_rec_ = 0.024). The rs1518108 polymorphism was associated with risk of hypertension (OR = 1.83, 95% CI: 1.16–2.86, *P*_cod1_ = 0.008 and OR = 1.68, 95% CI: 1.10–2.59, *P*_dom_ = 0.016), high levels of inflammation (hsCRP ≥ 3mg/L, OR = 1.73, 95% CI: 1.03–2.89, *P*_cod2_ = 0.037 and OR = 1.31, 95% CI: 1.01–1.70, *P*_rec_ = 0.036) and levels of TAT > p75 (OR = 1.69, 95% CI: 1.07–2.69, *P*_cod1_ = 0.025 and OR = 1.54, 95% CI: 1.004–2.37, *P*_dom_ = 0.048). In SA individuals, under different models, the rs1518108 polymorphism was associated with levels of GGT > p75 (OR = 0.51, 95% CI: 0.28–0.91, *P*_cod1_ = 0.023; OR = 0.35, 95% CI: 0.16–0.74, *P*_cod2_ = 0.006; OR = 0.46, 95% CI: 0.26–0.79, *P*_dom_ = 0.006 and OR = 0.58, 95% CI: 0.40–0.99, *P*_add_ = 0.004) and alkaline phosphatase (ALP) > p75 (OR = 0.46, 95% CI: 0.26–0.84, *P*_cod1_ = 0.011; OR = 0.48, 95% CI: 0.28–0.83, *P*_dom_ = 0.010 and OR = 0.68, 95% CI: 0.47–0.99, *P*_add_ = 0.046) (Table 4). In summary, in SA individuals, the rs1518108 was associated with high levels of GGT and ALP, whereas in controls, this polymorphism was associated with inflammation, hypertension and high levels of TAT. In controls, also the rs1400986 was associated with inflammation and low levels of GGT.

### 3.4. Association of the rs1400986 Genotypes with IL-20 Levels

Levels of IL-20 were determined in 106 control participants (34 with *CC*, 36 with *CT* and 36 with *TT* genotypes). Individuals with extreme outlier’s values were not included in the analysis (1 participant). SNP rs1400986 was associated with IL-20 levels. Figure 1 shows that individuals with *TT* genotype have significantly higher IL-20 levels than individuals with *CC + CT* genotypes (4.9 (3.1–10.7) pg/mL versus 3.6 (1.8–7.1) pg/mL respectively, *P* = 0.0313).

### 3.5. Correlation of rs1400986 Genotypes with IL-20 Levels and Adipose Tissue

As for body–mass index (BMI) and the VAT/SAT ratio, the individuals with *CC* genotypes showed a statistically significant negative correlation between these parameters (r^2^ = 0.021; *P* = 0.0087) and no such association was found in relation to *CT* + *TT* genotypes (*P* = 0.579) (Figure 2A). In order to better understand whether adipose tissue levels may impact the IL-20 concentration according to the genotypes of the rs1400986 polymorphism, we performed a correlation analysis in 106 control participants (*CC n* = 34, *CT n* = 36 and *TT n* = 36). Among individuals with the *CC* genotype, the IL-20 concentrations were negatively correlated with TAT (r^2^ = 0.23; *P* = 0.0037, Figure 2B), VAT (r^2^ = 0.12; *P* = 0.0427, Figure 2C) and SAT (r^2^ = 0.21; *P* = 0.006, Figure 2D). These correlations were not found in individuals with *CT* + *TT* genotypes. Thus, negative correlation between BMI and VAT/SAT ratio in individuals with rs1400986 *CC* genotype and among IL-20 levels and TAT, VAT and SAT was observed.

### 3.6. Haplotypes Analysis

The IL-20 polymorphisms were not in linkage disequilibrium (D’ > 0.048 and r^2^ > 0.09). In fact, 4 different haplotypes were observed; one of these haplotypes (*TT*) was significantly associated with a lower risk of SA (OR = 0.63, 95% CI: 0.50–0.80, *P* = 0.00016) (Table 5).

## 4. Discussion

The results of the study are the first demonstration of the association between *IL-20* polymorphisms and SA. Here, we provide genetic evidence that the rs1400986 and rs1518108 polymorphisms occurring in the *IL-20* gene were independently associated with a lower risk of developing SA. When these polymorphisms were analyzed as a haplotype, the association remaining significant. Associations with cardiovascular risk factors were observed in both studied groups. In AS individuals, the rs1518108 was associated with high levels of GGT and ALP, whereas in controls, this polymorphism was associated with high levels of hsCRP and TAT and with hypertension. In controls, also the rs1400986 was associated with high levels of hsCRP and low levels of GGT. High levels of IL-20 were observed in control individuals with rs1400986 *TT* genotype. Negative correlation between BMI and VAT/SAT ratio in individuals with rs1400986 *CC* genotype and among IL-20 levels and TAT, VAT and SAT was observed. Chen et al., [15] shown that IL-20 and its receptors are expressed in the human and experimental atherosclerosis plaque. More importantly, the authors demonstrated that systemic delivery of IL-20 accelerates atherogenesis in the apolipoprotein E-knockout mouse model. Our current study supports an important role of the *IL-20* polymorphisms in SA and in some cardiovascular risk factors.

It was found that IL20 cytokine has been involved in the developing of atherosclerosis [15,17], however, at the present, there are not studies that analyzed the association of the polymorphisms located in the gene that encodes this cytokine with the presence of atherosclerosis. The association of these polymorphisms with some inflammatory diseases has been reported, specifically in psoriasis and ulcerative colitis [28,29]. In African Americans, the rs1400986 and rs1518108 polymorphisms were associated with chronic hepatitis B infection [16]. Galimova et al., [30] analyzed 48 polymorphisms in 377 patients with psoriasis and 403 healthy controls and reported that the rs1400986 *T* allele was associated with decreased risk of psoriasis and that the combination of *IL10* rs1554286 and *IL20* rs1518108 was associated also with a reduced risk of presenting this disease. In these studies, were only compared the three different genotypes in each polymorphism, they did not use an analysis considering the inheritance models like in our study. In our study these polymorphisms were associated with lower risk of SA under different inheritance models. The rs1400986 was associated under the five models analyzed (cod1, cod2, dom, rec and add), whereas the rs1518108 was associated under two models (cod2 and add). This result corroborates the association, principally of the rs1400986 polymorphism with SA. An in-silico analysis showed that the rs1400986 polymorphism modify a binding site for the MZF1 transcriptional factor. MZF1 is a transcriptional regulator of several proteins; one of them is the SERPINA3 (serine proteinase inhibitor A3), protein that has been recently suggested as a potential predictive marker of clinical outcomes in myocardial infarction [31]. The association of the rs1400986 polymorphism with SA that we detected in our study could be related with the effect of the MZF2 transcriptional factor in the IL-20 levels.

In our study, the rs1518108 polymorphism was associated with high levels of GGT in SA individuals. High concentrations of GGT have been associated with cardiovascular diseases [32] and with non-alcoholic fatty liver disease (NAFLD) [33]. Is well known that fatty liver is an important condition associated with the developing of atherosclerosis [34]. In a previous study, increased expression of IL-20 was detected in obese patients with NAFLD [35].

Both IL-20 polymorphisms were associated with high levels of hsCRP in control individuals corroborating an inflammatory effect of this cytokine in these individuals. The rs1400986 *TT* genotype was associated with decreased risk of SA, however in control individuals this genotype was associated with high levels of IL-20, which is a pro-inflammatory cytokine. This contradictory result could be explained considering that the production of IL-20 and other molecules is a complex mechanism that involves not only changes at DNA level but also epigenetics modifications. Moreover, it is important, to considered that in our study, the levels of IL-20 were measured in circulating blood and only in a small group of control individuals. Unfortunately, these levels were not measured in SA individuals. The panel used for the determination of the IL-20 levels was designed by us and also included the IL-19 and IL-22 cytokines, both members of the IL-10 family. Associations of the levels of these cytokines with specific polymorphisms located in its respective genes are currently being analyzed.

It is important to mention, that the genotypes of *IL-20* rs1400986 polymorphism presented a negative correlation with TAT, VAT and SAT in control individuals. Thus, in individuals with rs1400986 *CC* genotype, the levels of the IL-20 cytokine decreased when the levels of TAT, VAT or SAT increase. Recently, Tanaka et al. [36] analyzed the impact of the VAT and SAT distribution on coronary plaque scores. They reported that high SAT and low VAT correlated inversely with the extent and severity of coronary artery plaques.

Our study presents some strengths: The group of individuals that we analyzed belong to the GEA project that was designed to examine the genetic bases of premature CAD and SA and its association with emerging and traditional cardiovascular risk factors in the Mexican population. This is a large cohort that was characterized from demographic, anthropometric, biochemical, clinical and tomographic point of view. Also, in the whole group of individuals, ancestry markers were determined and the distribution of Amerindian, European and African ancestry was similar in the groups included, thus the study has no ethnic bias [23]. Despite this, there are some limitations to consider. The results were not replicated in an independent set of patients and controls. Considering that this is the first time that polymorphisms located within the IL-20 gene are associated with SA and cardiovascular risk factors, replication in another cohort of patients is necessary to confirm these results. The IL-20 levels were only determined in a small sample of control individuals and not in the SA individuals. The study included only individuals with SA defined as CAC > 0.0, thus the association of these polymorphisms with coronary artery disease is mandatory.

## 5. Conclusions

In summary, our results indicate that the rs1400986 and rs1518108 polymorphisms in the *IL-20* gene were associated (independently or as a haplotype) with decreased risk of developing SA. These polymorphisms were also associated with some cardiovascular risk factors in individuals with SA and healthy controls. Healthy controls with the rs1400986 *TT* genotype presented high levels of IL-20. Negative correlation between BMI and VAT/SAT ratio in individuals with rs1400986 *CC* genotype and among IL-20 levels and TAT, VAT and SAT was observed. To the best of our knowledge, this is the first study that evaluates the association of *IL-20* polymorphisms with SA, cardiovascular risk factors and IL-20 levels. For this reason, the detected associations are not yet definitive and replicate studies in independent populations are warranted to confirm these findings.

## Figures and Tables

**Figure 1 biomolecules-10-00075-f001:**
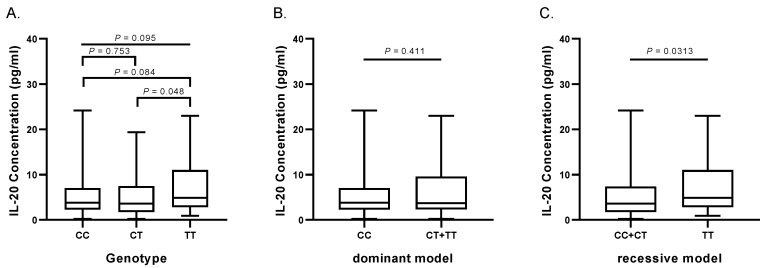
Association of the rs1400986 genotypes with IL-20 concentrations in control individuals. (**A**) Individuals with *TT* genotype have significantly higher IL-20 concentrations that those individuals with *CT* genotype (4.9 (3.1–10.7) pg/m vs. 3.8 (1.8–7.1) pg/mL, respectively, *P* = 0.048). (**B**) IL-20 concentrations in individuals with *CC* vs *CT + TT* genotypes (dominant model). No differences were observed. (**C**) IL-20 concentrations in individuals with *CC + CT* vs. *TT* genotypes (recessive model). Individuals with *TT* genotype have significantly higher IL-20 concentrations that those individuals with *CC + CT* genotype (4.9 (3.1–10.7) pg/m vs. 3.6 (1.8–7.1) pg/mL, respectively, *P* = 0.0313). Lines into bars indicate median; interquartile range (IQR 25–75) is shown in graphic representation. *P*: Kruskal-Wallis test and Mann-Whitney *U-*test.

**Figure 2 biomolecules-10-00075-f002:**
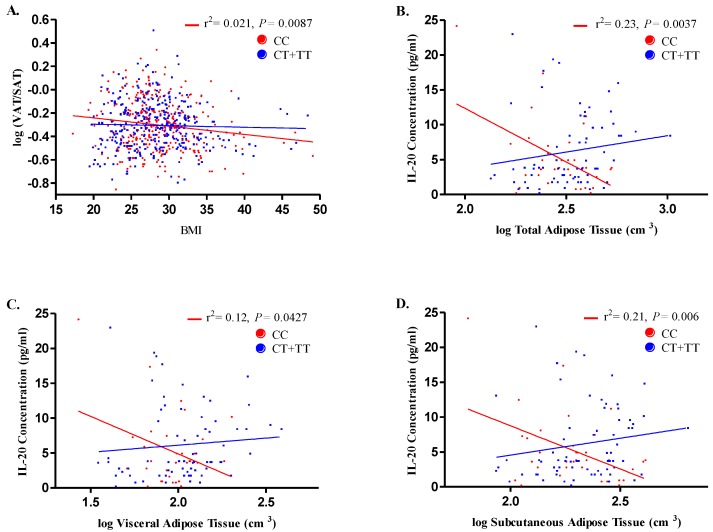
Correlation between adipose tissue distributions, body mass index and IL-20 concentrations according to rs1400986 genotypes in control individuals. Lines represent simple linear regression: blue lines represent *CT + TT* genotypes carriers and red represent *CC* genotype carriers. (**A**) Overall, body mass index (BMI) was negatively correlated with visceral to subcutaneous adipose tissue ratio (VAT/SAT) in individuals with *CC* genotypes but not in individuals with *CT + TT* genotypes. On adipose tissue stratification, a negatively and significant correlation of IL-20 levels and log TAT (**B**), log VAT (**C**) and log SAT (**D**) was observed in individuals with *CC* genotypes.

**Table 1 biomolecules-10-00075-t001:** Clinical and metabolic characteristics of the studied groups.

	Control (*n* = 672)	SA (*n* = 274)	*p*
Age (years)	52 ± 9	59 ± 8	<0.0001
Gender (% male)	38.2	72.6	<0.0001
Body Mass Index (kg/m^2^)	27.9 (25.4–30.9)	28.1 (25.6–31.3)	0.219
Waist Circumferences (cm)	93.4 ± 11.7	97.4 ± 11.1	<0.0001
Systolic Blood Pressure (mmHg)	115 (106–126)	124 (113–137)	<0.0001
Diastolic Blood Pressure (mmHg)	72 (66–78)	77 (70–83)	<0.0001
Total Adipose Fat (cm^3^)	443 (350–542)	442 (353–569)	0.391
Visceral Adipose fat (cm^3^)	146 (105–188)	180 (141–230)	<0.0001
Subcutaneous Adipose fat (cm^3^)	286 (218–371)	260 (193–340)	0.002
Total Cholesterol (mg/dL)	190 (168–209)	198 (171–221)	0.002
HDL-C (mg/dL)	47 (37–57)	43 (36–50)	<0.0001
LDL-C (mg/dL)	115.6 (96.2–133.2)	124.4 (102.3–145.2)	<0.0001
Triglycerides (mg/dL)	141 (107–194)	158 (118–206)	0.007
Non-HDL-Cholesterol (mg/dL)	140 (120–162)	153 (129–175)	<0.0001
ALT (IU/L)	23 (17–32)	22 (17–32)	0.819
AST (IU/L)	25 (21–30)	25 (21–31)	0.542
GGT (IU/L)	24 (17–41)	29 (21–41)	0.001
Alkaline Phosphatase (IU/L)	81 (68–98)	77 (65–93)	0.013
Apo B (mg/dL)	86 (72–106)	96 (79–119)	<0.0001
Apo A1 (mg/dL)	132 (113–157)	133 (113–157)	0.81
Apo-B/Apo-A	0.65 (0.51–0.84)	0.7 (0.6–0.9)	0.001
Glucose (mg/dL)	90 (84–97)	95 (87–107)	<0.0001
Insulin (µIU/mL)	18 (13–24)	19 (13–25)	0.212
HOMA-IR	3.9 (2.7–5.8)	4.7 (3.1–6.8)	<0.0001
hsCRP (mg/dL)	1.69 (0.87–3.46)	1.71 (0.89–3.45)	0.76
Creatinine (mg/dL)	0.8 (0.7–0.9)	0.9 (0.7–1.1)	<0.0001
Adiponectin (µg/mL)	8.4 (5–12.9)	6.4 (4.2–10.2)	<0.0001
Uric Acid (mg/dL)	5.4 (4.4–6.4)	5.9 (4.9–6.9)	<0.0001
Albumin (µg/mL)	6.3 (2.9–12)	7.2 (2.9–19)	0.026
Free Fatty Acid (mEq/L)	0.5 (0.4–0.7)	0.6 (0.4–0.7)	0.631
IR of the Adipose Tissue	9.7 (6.2–14.5)	10.2 (6.6–13.9)	0.629

Data are shown as mean ±SD, median (interquartile range) or percentage. Comparisons were made using Student’s t-test or Mann-Whitney U test, as appropriate, for continuous variables and by Chi square analysis for categorical variables. SA: Subclinical atherosclerosis; IR: Insulin resistance; hsCRP: High sensitivity C reactive protein; HOMA: Homeostasis model assessment of insulin resistance; GGT: Gamma Glutamyl transpeptidase; AST: Aspartate aminotransferase; HDL-C: High density lipoprotein-cholesterol; LDL-C: Low density lipoprotein-cholesterol; ALT: Alanine aminotransferase.

**Table 2 biomolecules-10-00075-t002:** Cardiovascular factors prevalence in the study population.

	Control (*n* = 672)	SA (*n* = 274)	* *P*
Total Cholesterol >200 mg/dL (%)	35.4	47.1	0.001
LDL-Cholesterol > 130 mg/dL (%)	29.3	42.9	<0.0001
Hypoalphalipoproteinemia (%)	47.7	45.3	0.518
Hypertriglyceridemia (%)	45.2	53.5	0.022
Non-HDL-Cholesterol > 160 mg/dL (%)	25.9	42	<0.0001
Overweight (%)	45.8	47.4	0.114
Obesity (%)	31.1	33.9	0.083
Abdominal Obesity (%)	79.6	82.1	0.417
Type 2 Diabetes Mellitus (%)	10.6	23	<0.0001
Hyperinsulinemia (%)	55.4	62.8	0.023
Insulin resistance (%)	57.5	67.9	0.002
Metabolic Syndrome (%)	40.6	54	<0.0001
Hypertension (%)	29.2	49.6	<0.0001
High Total Abdominal Tissue (%)	55.4	61.7	0.045
High Subcutaneous Abdominal Tissue (%)	50	54.7	0.106
High Visceral Abdominal Tissue (%)	58.8	73.7	<0.0001
Fatty Liver (%)	32.1	39.3	0.024

Data is shown as percentage. * Comparisons were made using Chi square analysis. SA: Subclinical atherosclerosis, LDL: Low density lipoprotein and HDL: High density lipoprotein.

**Table 3 biomolecules-10-00075-t003:** Association between *IL-20* gene polymorphisms and subclinical atherosclerosis.

SNP	Model	Genotypes and Alleles	SA	Control	*p*	OR	95% CI
			*n*	*n*			
rs1400986		*CC*	177	336			
		*CT*	87	293			
		*TT*	10	43			
		*C*	441	965	0.0001	0.61	0.48–0.78
		*T*	107	379			
	codominant1	*CC*	177	336	0.0001	0.51	0.36–0.73
		*CT*	87	293			
	codominant2	*CC*	177	336	0.014	0.36	0.16–0.81
		*TT*	10	43			
	dominant	*CC*	177	336	0.0001	0.49	0.35–0.69
		*CT + TT*	97	336			
	recessive	*CC + CT*	264	629	0.063	0.47	0.21–1.04
		*TT*	10	43			
	additive	–	–	–	0.0001	0.55	0.41–0.73
rs1518108		*CC*	79	181			
		*CT*	140	336			
		*TT*	55	155			
		*C*	298	698	0.229	0.89	0.75–1.06
		*T*	250	646			
	codominant1	*CC*	79	181	0.246	0.79	0.54–1.16
		*CT*	140	336			
	codominant2	*CC*	79	181	0.048	0.62	0.39–0.99
		*TT*	55	155			
	dominant	*CC*	79	181	0.102	0.74	0.51–1.06
		*CT + TT*	195	491			
	recessive	*CC + CT*	219	517	0.110	0.72	0.49–1.07
		*TT*	55	155			
	additive	–	–	–	0.0480	0.79	0.63–0.99

Models were adjusted for age, gender, body mass index, current smoking status, alanine aminotransferase, aspartate aminotransferase and uric acid. SA: Subclinical atherosclerosis.

**Table 4 biomolecules-10-00075-t004:** Association among *IL-20* gene polymorphisms and cardiovascular risk factors in controls and SA individuals.

SNP	Model	Genotypes	Variable		*p*	OR	95% CI
(i) Controls							
rs1400986			Inflammation				
			Yes *n* = 203	No *n* = 469			
	codominant1	*CC*	91	245	0.047	1.45	1.01–2.10
		*CT*	101	192			
			GGT > 75				
			Yes *n* = 267	No *n* = 400			
	codominant2	*CC*	140	196	0.023	0.41	0.19–0.88
		*TT*	10	32			
	recessive	*CC + CT*	257	368	0.024	0.42	0.19–0.89
		*TT*	10	32			
rs1518108			Hypertension				
			Yes *n* = 196	No *n* = 476			
	codominant1	*CC*	38	143	0.008	1.83	1.16–2.86
		*CT*	113	223			
	dominant	*CC*	38	143	0.016	1.68	1.10–2.59
		*CT + TT*	158	333			
			Inflammation				
			Yes *n* = 203	No *n* = 469			
	codominant2	*CC*	43	138	0.037	1.73	1.03–2.89
		*TT*	51	104			
	recessive	*CC + CT*	152	365	0.036	1.31	1.01–1.70
		*TT*	51	104			
			Total abdominal tissue >75				
			Yes *n* = 360	No *n* = 290			
	codominant1	*CC*	96	79	0.025	1.69	1.07–2.69
		*CT*	179	146			
	dominant	*CC*	96	79	0.048	1.54	1.004–2.37
		*CT + TT*	264	211			
(ii) SA							
rs1518108			GGT > 75				
			Yes *n* = 111	No *n* = 163			
	codominant1	*CC*	43	36	0.023	0.51	0.28–0.91
		*CT*	51	88			
	codominant2	*CC*	43	36	0.006	0.35	0.16–0.74
		*TT*	17	39			
	dominant	*CC*	43	36	0.006	0.46	0.26–0.79
		*CT + TT*	68	127			
	additive	*-*	-	-	0.004	0.58	0.40–0.99
			ALP > 75				
			Yes *n* = 91	No *n* = 183			
	codominant1	*CC*	36	43	0.011	0.46	0.26–0.84
		*CT*	38	101			
	dominant	*CC*	36	43	0.01	0.48	0.28–0.83
		*CT + TT*	55	140			
	additive	*-*	-	-	0.046	0.68	0.47–0.99

Table shows the model with significant associations. Models were adjusted for age, gender and body mass index. Inflammation was considered when hsCRP ≥ 3mg/L.

**Table 5 biomolecules-10-00075-t005:** *IL-20* haplotype frequencies in SA and healthy controls.

Haplotypes	SA (*n* = 274)		Control (*n* = 672)		*Χ* ^2^	*P*	OR	95% CI
	*n*	%	*n*	%				
*CC*	134	48.9	298	44.4	3.88	0.077	1.19	1.01–1.42
*CT*	87	31.6	184	27.4	3.376	0.066	1.22	1.00–1.48
*TT*	39	14.2	139	20.7	10.684	0.00016	0.63	0.50–0.80
*TC*	14	5.4	51	7.6	2.94	0.086	0.69	0.48–0.99

The order of the alleles in the haplotypes is according to the positions of the polymorphisms in the chromosome (rs1400986 and rs1518108). SA: subclinical atherosclerosis; OR: odds ratio.

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
