# Peer review of "Common Variants in IL-20 Gene are Associated with Subclinical Atherosclerosis, Cardiovascular Risk Factors and IL-20 Levels in the Cohort of the Genetics of Atherosclerotic Disease (GEA) Mexican Study"

_biomolecules, 2020, doi:10.3390/biom10010075_

Round 1
Reviewer 1 Report
The authors investigated the association of IL-20 gene polymorphism with subclinical atherosclerosis as well as the IL-20 level in normal donors. The study is interesting but could be further improved. Please see below for some concerns.
The title should be one sentence only. Raw data for individual donors should be provided in the supplemental materials. As indicated in Table 1, the variation of gender between Control and SA group is very high (38.2% vs 72.6%) and should be considered in this study. The authors may have to compare the female and male samples separately. Why the authors only analyzed the polymorphism of rs1400986 and rs1518108 in this study? Figure 1, the authors should show separated columns for CC, TT and CT . Also, how is IL-20 level in SA donors with different SNPs? Table 5, what’s the difference between “CT” and “TC”?Author Response
1.- The title should be one sentence only.
Answer:The title has been modify.
New title: Common variants in IL-20 gene are associated with subclinical atherosclerosis, cardiovascular risk factors and IL-20 levels in the cohort of the GEA Mexican Study.
2.- Raw data for individual donors should be provided in the supplemental materials.
Answer:The raw data has been included in the suplemental material.
3.- As indicated in Table 1, the variation of gender between Control and SA group is very high (38.2% vs 72.6%) and should be considered in this study. The authors may have to compare the female and male samples separately.
Answer:As was suggested by the reviewer, female and men individuals has been analysed separately. The results obtained in this analysis is similar to that in the whole group of individuals. The rs1400986 polymorphisms was associated with SA in both groups, whereas the rs1518108 polymorphism was associated with SA in the group of men. We considered that this results should be take with care because the numbre of individuals included is low. An study in a large cohort of female and men could help to define if the associations detected are different considering the sex of the individuals. A table with this analysis has been included in Supplementary Material.
The phrase “Considering the variation of gender between controls and SA individuals, these groups were analyzed separately. The results obtained in this analysis are similar to that in the whole group of individuals. The rs1400986 polymorphism was associated with SA in both groups, whereas the rs1518108 polymorphism was associated with SA in the group of men (Table 1. Supplental Material). This analysis should be taken with care because the number of individuals included is low.”has been added in the results section.
4.- Why the authors only analyzed the polymorphism of rs1400986 and rs1518108 in this study?
Answer: These two polymorphisms were analyzed because both of them have been associated with other inflammatory disease, such as, psoriasis and ulcerative colitis. Also, they were associated with chronic hepatitis B infection in African-Americans. On the other hand, an in silico analysys that we made showed that the rs1400986 polymorphism modify a binding site for the MZF1 transcriptional factor, having a possible functional effect. In order to clarify this point, the phrase “The IL-20gene is located in chromosome 1; some of its polymorphisms have been associated with psoriasis and ulcerative colitis [14,15].”has been changed to “The IL-20gene is located in chromosome 1, and two polymorphisms (rs1400986 and rs1518108) have been associated with inflammatory diseases, such as, psoriasis and ulcerative colitis [14,15]. Also, these polymorphisms were associated with chronic hepatitis B infection in African-Americans. On the other hand, an “in silico” analysis that we made showed that the rs1400986 polymorphism modify a binding site for the MZF1 transcriptional factor, having a possible functional effect. Despite the important role of this cytokine in the inflammatory process and in consequence in the development of atherosclerosis [15,17],at the present, there are not studies that analyzed the association of the polymorphisms located in the gene that encodes this cytokine with the presence of atherosclerosis and cardiovascular risk factors.Thus, the aim of the present study was to evaluate the association of rs1400986, and rs1518108 IL-20polymorphisms with SA, and cardiovascular risk factors in a Mexican population.” in the introduction section.
5.- Figure 1, the authors should show separated columns for CC, TT and CT.
Answer:As was suggested by the reviewer, the levels of IL-20 are showed separatelly in the different genotypes (CC, CT, and TT). In the same figure, the levels of IL-20 in carriers of CC vs CT+TT (dominant model) and carriers of CC+CT vs TT (recessive model) are showed.
6.- Also, how is IL-20 level in SA donors with different SNPs?
Answer: Unfortunately, the IL-20 levels were not determined in SA individuals. This is a limitation that has been added in the discussion. The phrase "The IL-20 levels were only determined in a small sample of control individuals and not in the SA individuals" has been added.
7.- Table 5, what’s the difference between “CT” and “TC”?
Answer:The difference is the allele in each position. CT(C for rs1400986 and Tfor rs1518108 ), whereas TC(Tfor rs1400986 and Cfor rs1518108). In order to clarify this point, the footnote of the table 5 has been modified. “The order of the alleles in the haplotypes is according to the positions of the polymorphisms in the chromosome (rs1400986 and rs1518108).
Reviewer 2 Report
Manuscript ID: Biomolecules-666025
Type of manuscript: Article
Title: Common variants in IL-20 gene are associated with subclinical atherosclerosis, fat distribution and metabolic traits. Genetics of Atherosclerosis Disease Study.
Researchers of the manuscript entitled “Common variants in IL-20 gene are associated with subclinical atherosclerosis, fat distribution and metabolic traits. Genetics of Atherosclerosis Disease Study" reported that the two common variants of pro-inflammatory IL20 gene are also associated with subclinical atherosclerosis in examined Mexicans population.
The work is written carelessly.
The title does not inform us that only applies to the Mexicans population.
There is no relation to the distribution of fat combined with genetic variants of the IL20 gene in the results. The words "metabolic traits" are not used in the text, but only in the title.
Title, aim of work and conclusions have to be consistent!
There is no conclusion in the abstract. Last sentence is not appropriate.
In the introduction
- there is no information about subclinical atherosclerosis, what parameters indicate it,
- line 41 part of sentence need correction
- VEGF stands for vascular endothelial growth factor,
- is there evidence for the theory "in consequence in the development of atherosclerosis"?
In the materials and methods
There is lack method of IL20 level estimation.
Line 80 “…was isolated by standard techniques.” You have to write exact citation.
Line 81 ” … genotyping assay…” You need to write more about the reagents which you are used.
Line 88 “calculated as the case may be.” What did you mean?
Line 95 Can you complete the meaning of individual models here? It will be difficult for the reader to look for it to reference 21.
Line 101 In the results (Table 3) you write E H-W and P with a subscript E H-W then make it common.
Line 102 “version 4:1” you can write v. 4.1. or version 4.1
Line 103 Does Haploview software have a citation?
In the results
Can you check the hsCRP values in Table 1 ? Is everything right?
When explaining the abbreviations under the table, you start with a capital letter once and another with a lower case - take one way.
Line 133 Write P with a subscript E H-W.
Table 3 should be reformatted, because is not readable.
There are no explanations for the abbreviations. I observe their lack if going of genotypes, models (Tables 3 and 4) and haplotypes. Genotypes and haplotypes require improvement in all work !!! Write exactly!
Reference19 is not available for most readers. It will be difficult to look for explanations of these abbreviations in reference 21st, they should be written.
If you describe model abbreviations in materials and methods, you can use them in lines 136 -141 and 149-156.
Table 4 must be consistent with the title and requires reformatting , because is not readable.
Figure 1 n=106 for each or genotype CC+CT n=70 and TT n=36 ?
Change caption of the Figure 1. Like Comparison ……
Second part of the caption was repeated.
What does this mean for TT control individuals that they have higher IL20 levels?
VAT/SAT ratio and BMI, and IL20 concentration and TAT, VAT, SAT negative correlations, what does this mean for CC? Explain.
Check the correlation description in chapter 3.5.
Table 5 – Haplotype require improvement - complete. n=?
In discussion
Sentence in line 201 – 202 In which results I can observed this? Did you describe it in the results?
Sentence in line 203 Is it true?
Line 205 For the first time you used abbreviation ALP ! What is mean? Applicate earlier!
Line 201 - 210 includes results. There are no such summarizing sentences in the results.
Line 216 sentence need reconstruction and reference.
There is a lot of speculation in the discussion. Verify that. Take into consideration used different models. You described them in the results and in the abstract , for what? maybe it's important?
Why do you not include them to the discussion? May be, you can compare them to other pathologies.
Literature/references need correction according to the Journal requirements.
The manuscript needs accurately read and language correction.
Author Response
1.- The work is written carelessly.
Answer:The language has been revised carefully.
2.- The title does not inform us that only applies to the Mexicans population.
Answer:The title has been modify as was suggested by the reviewer 1 and now include the term “cohort of the GEA Mexican Study.”.
New title: Common variants in IL-20 gene are associated with subclinical atherosclerosis, cardiovascular risk factors and IL-20 levels in the cohort of the GEA Mexican Study.
3.- There is no relation to the distribution of fat combined with genetic variants of the IL20 gene in the results. The words "metabolic traits" are not used in the text, but only in the title.
Answer: As was suggested by the reviewer, the title has been modified and the term “metabolic traits” has been deleted.
4.- Title, aim of work and conclusions have to be consistent!
Answer:The title and objective have been modified. Now they are according with the conclusions.
New title: Common variants in IL-20 gene are associated with subclinical atherosclerosis, cardiovascular risk factors and IL-20 levels in the cohort of the GEA Mexican Study.
The objetive has been modified “The aim of the study was to evaluate the association of two IL-20polymorphisms (rs1400986 and rs1518108) with subclinical atherosclerosis (SA), cardiovascular risk factors, and IL-20 levels in a cohort of Mexican individuals.”
5.- There is no conclusion in the abstract. Last sentence is not appropriate.
Answer:We agree with reviewer, the last sentence in the abstract has been deleted. As conclusion we included the follow sentence “Our results indicate that the IL-20 rs1400986 and rs1518108 polymorphisms were associated with decreased risk of developing SA, and with some cardiovascular risk factors in individuals with SA, and healthy controls. Negative correlation between BMI and VAT/SAT ratio in individuals with rs1400986 CCgenotype, and among IL-20 levels and TAT, VAT and SAT was observed.”
In the introduction
1.- there is no information about subclinical atherosclerosis, what parameters indicate it,
Answer: In the introduction information about the subclinical atehrosclerosis has been included. The phrase “SA develops over several decades and often remains asymptomatic until the occurrence of an acute, life-threatening event. Two subclinical measures of atherosclerosis have been used to predict CAD. One is carotid intima-media thickness (cIMT), a measure of the intimal and medial layers of the carotid artery walls, and the other is coronary artery calcification (CAC), a marker of subclinical coronary atherosclerosis [5,6].”has been included. Two references have been included.
Newman AB, Naydeck BL, Sutton-Tyrrell K, Edmundowicz D, O’Leary D, Kronmal R, et al.Relationship between coronary artery calcification and other measures of subclinical cardiovascular disease in older adults. Arterioscler Thromb Vase Biol 2002:22:1674–9.
Wagenknecht LE, Langefeld CD, Carr JJ, Riley W, Freedman BI, Moossavi S, et al. Race-specific relationships between coronary and carotid artery calcification and carotid intimal medial thickness. Stroke2004:35:e97–9. Doi: 10.1161/01.STR.0000127081.99767.1d.
2.- - line 41 part of sentence need correction
Answer:The phrase “Visceral adipose tissue (VAT) accumulation is clearly associated with a higher risk of T2DM and CAD and is positively associated with cardiometabolic risk factors, such as blood pressure, lipids and measures of glucose homeostasis [5–7].” has been changed to “Visceral adipose tissue (VAT) accumulation is clearly associated with a higher risk of T2DM and CAD and is positively associated with cardiovascular risk factors [5–7].”
3.- VEGF stands for vascular endothelial growth factor,
Answer:Yes, the complete term has been included
4.- is there evidence for the theory "in consequence in the development of atherosclerosis"?
Answer:There are some studies that suggest the participation of the IL-20 cytokine in the develpment of atehrosclerosis. Two references with this information have been included in the introduction section.
Caligiuri G, Kaveri SV, Nicoletti A. IL-20 and atherosclerosis: another brick in the wall.Arterioscler Thromb Vasc Biol. 2006 Sep;26(9):1929-30.
Chen WY, Cheng BC, Jiang MJ, Hsieh MY, Chang MS. IL-20 is expressed in atherosclerosis plaques and promotes atherosclerosis in apolipoprotein E-deficient mice.Arterioscler Thromb Vasc Biol. 2006 Sep;26(9):2090-5.
In the materials and methods
1.- There is lack method of IL20 level estimation.
Answer:The Il-20 levels were determined by Luminex. The phrase “In a subsample of 106 control individuals, IL-20 plasma concentrations were quantified using a Bioplex system (Bio-Rad, Contra Costa County, State of California, USA). The data were analyzed using Bio-Plex Manager software. Data were expressed in pg/mL.”in material and methods section.
2.- Line 80 “…was isolated by standard techniques.” You have to write exact citation.
Answer:The techinique used for the DNA extraction has been included. The phrase “Genomic DNA from whole blood containing EDTA was isolated with a no enzymatic method [Lahiri & Nurnberger, 1991].” has been added.
The reference “Lahiri, D. K., & Nurnberger, J. I. Jr (1991). A rapid non‐enzymatic method for the preparation of HMW DNA from blood for RFLP studies. Nucleic Acids Research, 19, 5444.” has been included.
3.- Line 81 ” … genotyping assay…” You need to write more about the reagents which you are used.
Answer:In response to the reviewer, the phrase“The 5’ exonuclease TaqMan genotyping assays were used to determine the IL-20(rs1400986 and rs1518108) polymorphisms. The determinations were made on an ABI Prism 7900HT Fast Real-Time PCR system, according to manufacturer’s instructions (Applied Biosystems, Foster City, CA, USA).” was changed to“According to the manufacturer’s instructions (Applied Biosystems, Foster City, CA, United States), the rs1400986 (C___1747382_10),and rs1518108 (C___1747381_10)IL-20polymorphisms were determined in genomic DNA using 5′ exonuclease TaqMan genotyping assays on an ABI Prism 7900HT Fast Real-Time PCR system.”in material and methods section.
4.- Line 88 “calculated as the case may be.” What did you mean?
Answer:The phrase “Numerical variables were expressed as means +standard deviation. Medians, interquartile ranges, and frequencies were calculated as the case may be.”was changed to “Data are expressed as the mean (standard deviation), median (interquartile range) or frequencies.”
5.- Line 95 Can you complete the meaning of individual models here? It will be difficult for the reader to look for it to reference 21.
Answer:The meaning of the models and its abbreviations have been included. The phrase “The polymorphisms associations with subclinical atherosclerosis and other variables were analyzed using logistic regression under the following inheritance models: additive, codominant 1, codominant 2, dominant, and recessive.”was changed to “The assocations of the polymorphisms with SA and cardiovascular risk factors were analyzed using logistic regression under the following inheritance models: additive (add-major allele homozygotes vs. heterozygotes vs. minor allele homozygotes), codominant 1 (cod1-major allele homozygotes vs. heterozygotes), codominant 2 (cod2-major allele homozygotes vs. minor allele homozygotes), dominant (dom-major allele homozygotes vs. heterozygotes+minor allele homozygotes), and recessive (rec-major allele homozygotes+heterozygotes vs. minor allele homozygotes). When the association with SA was tested, the models were adjusted for age, gender, body mass index, current smoking status, alanine aminotransferase, aspartate aminotransferase, and uric acid. To evaluate the association with cardiovascular risk factors, the models were adjusted for age, gender and BMI. ”
6.- Line 101 In the results (Table 3) you write E H-W and P with a subscript E H-W then make it common.
Answer:In the new table, the value of HWE was not included.
7.- Line 102 “version 4:1” you can write v. 4.1. or version 4.1
Answer:This has been corrected. The correct term is “version 4.1”
8.- Line 103 Does Haploview software have a citation?
Answer:The Haploview software does not have a citation.
In the results
1.- Can you check the hsCRP values in Table 1 ? Is everything right?
Answer:The values have been corrected
2.- When explaining the abbreviations under the table, you start with a capital letter once and another with a lower case - take one way.
Answer:This has been corrected. We used the capital letter in all cases.
3.- Line 133 Write P with a subscript E H-W.
Answer:This has been corrected, we use the abbreviation HWE.
4.- Table 3 should be reformatted, because is not readable.
Answer:The table has been reformatted.
5.- There are no explanations for the abbreviations. I observe their lack if going of genotypes, models (Tables 3 and 4) and haplotypes. Genotypes and haplotypes require improvement in all work !!! Write exactly!
Answer: The tables have been reformatted and now included the alleles and genotypes. The models are included. The abbreviations of the models were included in material and methods, then in the text, we only inlclude the abbreviation.
6.- Reference19 is not available for most readers. It will be difficult to look for explanations of these abbreviations in reference 21st, they should be written.
Answer:The abbreviations included in the reference 19 have been written in the text.
7.- If you describe model abbreviations in materials and methods, you can use them in lines 136 -141 and 149-156.
Answer:The abbreviations of the models were included in material and methods, then in the text, we use the abbreviations.
8.- Table 4 must be consistent with the title and requires reformatting , because is not readable.
Answer: The title of the table has been modified and reformatting.
New table title: Association among IL-20gene polymorphisms and cardiovascular risk factors in controls and SA individuals.
9.- Figure 1 n=106 for each or genotype CC+CT n=70 and TT n=36 ?
Answer: This is correct. CCn=34, CT=36, and TTn=36. This information has been included in thechapter 3.4.
10.- Change caption of the Figure 1. Like Comparison ……
Answer:The caption has been modify.
Association of the rs1400986 genotypes with IL-20 concentrations in control individuals.
11.- Second part of the caption was repeated.
Answer: This has been corrected.
12.- What does this mean for TT control individuals that they have higher IL20 levels?
Answer: In this case, the rs1400986 TT genotype was asociated with decreased risk of SA, however in control individuals this genotype was associated with high levels of IL-20. This controvertian result is discussed and the phrase “In a small sample of control individuals, we detected those carriers of rs1400986 TTgenotype presented high levels of IL-20 when compared to carriers of CC+CTgenotypes, suggesting that this polymorphism could be involved in the regulation of the levels of this cytokine.”has been changed to“The rs1400986 TTgenotype was associated with decreased risk of SA, however in control individuals this genotype was associated with high levels of IL-20, which is a pro-inflammatory cytokine. This contradictory result could be explained considering that the production of IL-20 and other molecules is a complex mechanism that involves not only changes at DNA level but also epigenetics modifications. Moreover, it is important to considered that in our study, the levels of IL-20 were measured in circulation and only in a small group of control individuals. Unfortunately, these levels were not measured in SA individuals.”has been included in the discussion section
13.- VAT/SAT ratio and BMI, and IL20 concentration and TAT, VAT, SAT negative correlations, what does this mean for CC? Explain.
Answer: Thus, in individuals with rs1400986 CC genotype, the levels of the IL-20 cytokine decreased when the levels of TAT, VAT or SAT increase. This phrase has been added in the discussion section.
14.- Check the correlation description in chapter 3.5.
Answer: In order to clarify the description, the phrase“As for BMI and the VAT/SAT ratio, while in individuals with CCgenotypes BMI showed a statistically significant negative correlation with the VAT/SAT (r2 = 0.021; P = 0.0087), no association was found in individuals with CTor TTgenotypes (P = 0.579) (Figure 2A). In order to better understand whether adipose tissue levels may impact the IL-20 concentration, we performed a correlation analysis in 106 control participants (CCn = 34, CTn = 36 and TTn= 36). Among individuals with the CCgenotype, the IL-20 concentrations were negatively correlated with total adipose tissue (r2= 0.23; P = 0.0037, Fig 2B.), visceral adipose tissue (r2 = 0.12; P = 0.042 Fig 2C.) and subcutaneous adipose tissue (r2 = 0.21; P = 0.0060, Fig 2D.). These correlations were not found in individuals with CTor TTgenotypes.”has been changed to “As for BMI and the VAT/SAT ratio, while in individuals with CCgenotypes BMI showed a statistically significant negative correlation with the VAT/SAT (r2 = 0.021; P = 0.0087), no association was found in individuals with CT+TTgenotypes (P = 0.579) (Figure 2A).In order to better understand whether adipose tissue levels may impact the IL-20 concentration according to the genotypes of the rs1400986 polymorphism, we performed a correlation analysis in 106 control participants (CCn = 34, CTn = 36 and TTn= 36). Among individuals with the CCgenotype, the IL-20 concentrations were negatively correlated with TAT (r2= 0.23; P = 0.0037, Fig 2B), VAT (r2 = 0.12; P = 0.0427 Fig 2C) and SAT (r2= 0.21; P = 0.006, Fig 2D). These correlations were not found in individuals with CT+TTgenotypes.”
15.- Table 5 – Haplotype require improvement - complete. n=?
Answer:The table 5 has been impovement and the numbers have been included.
In discussion
1.- Sentence in line 201 – 202 In which results I can observed this? Did you describe it in the results?
Answer: In order to clarify this point, the phrase“Here, we provide genetic evidence that minor allele Tof rs1400986 and rs1518108 in the IL-20gene were independently associated with a lower risk of developing SA.”has been changed to “Here, we provide genetic evidence that the rs1400986 and rs1518108 polymorphisms in the IL-20gene were independently associated with a lower risk of developing SA.”
2.- Sentence in line 203 Is it true?
Answer: Considering that the association detected for the TT haplotype is only stronger than the association obtained with the rs1518108 polymorphism, but not for the association detected for the rs1400986, the phrase“When these polymorphisms were analyzed as a haplotype, the association was stronger.” has been changed to “When these polymorphisms were analyzed as a haplotype, the association remaining significant.”
3.- Line 205 For the first time you used abbreviation ALP ! What is mean? Applicate earlier!
Answer:The abbreviations was defined in results, section 3.3..
ALP (alkaline phosphatase).
4.- Line 201 - 210 includes results. There are no such summarizing sentences in the results.
Answer:In order to clarify this point, a summarizing sentences have been added in the results section.
In section 3.2, the phrases “Thus, both polymorphisms were independently associated with a lower risk of developing SA.”has been added.
In section 3.3, the phrase “In summary, in SA individuals, the rs1518108 was associated with high levels of GGT and ALP, whereas in controls, this polymorphism was associated with inflammation, hypertension, and high levels of TAT. In controls, also the rs1400986 was associated with inflammation and low levels of GGT.”has been added.
In section 3.5, the phrase “Thus, negative correlation between BMI and VAT/SAT ratio in individuals with rs1400986 CCgenotype and among IL-20 levels and TAT, VAT and SAT was observed.”has been added.
5.- Line 216 sentence need reconstruction and reference.
Answer:The sentense “In spite that the IL20 cytokine has been involved in the developing of atherosclerosis, at the present, there are not studies that analyzed the association of the polymorphisms located in the IL-20gene with the presence of atherosclerosis”.has been changed to “IL20 cytokine has been involved in the developing of atherosclerosis [15,17], however, at the present, there are not studies that analyzed the association of the polymorphisms located in the gene that encodes this cytokine with the presence of atherosclerosis.”
6.- There is a lot of speculation in the discussion. Verify that. Take into consideration used different models. You described them in the results and in the abstract , for what? maybe it's important?
Why do you not include them to the discussion? May be, you can compare them to other pathologies.
Answer: As suggested the reviewer, the different models analysed in our study have been considered in the discussion section. Some phrases have been included.
The phrases“In these studies, were only compared the three different genotypes in each polymorphisms, they did not use an analysis considering the inheritance models like in our study. In our study these polymorphisms were associated with lower risk of SA under different inheritance models. The rs1400986 was associated under the five models analyzed (cod1, cod2, dom, rec, and add), whereas the rs1518108 was associated under 2 models (cod2 and add). This result corroborates the association, principally of the rs1400986 polymorphism with SA”
7.- Literature/references need correction according to the Journal requirements.
Answer: The references have been corrected according to the journal requirements. The format was revised in the journal and also a recent published paper in the journal was revised
8.- The manuscript needs accurately read and language correction.
Answer:The language has been revised carefully.
Round 2
Reviewer 1 Report
The authors addressed most of my concerns. However, as mentioned previously in point 2, raw data is necessary to be included for the validity of the data. Regarding the IL-20 level shown in Figure 1, it's still not clear how the IL-20 is quantified and how the original data looks like. In the methods part, the authors wrote "In a subsample of 106 control individuals, IL-20 plasma concentrations were quantified using a 94 Bioplex system (Bio-Rad, Contra Costa County, State of California, USA)." It's important to know which kit is exactly used for this experiment and the authors should provided the kit catalog number as well as describing the protocol. The related raw data should be included in the manuscript. If other cytokine levels are also measured in the same experiments, the authors should also briefly discussed.
Author Response
Answer:Raw data with the IL-20 levels in the 106 healthy controls was included as supplemental material.
The panel used for the determinations was designed for us, and we included other members of the IL-10 family (IL-19, and 22). This point is comented in the discussion section, and the phrase “The panel used for the determination of the IL-20 levels was designed by us, and also included the IL-19, and IL-22 cytokines, both members of the IL-10 family. Associations of the levels of these cytokines with specific polymorphisms located in its respective genes are currently being analyzed.”has been added in the discussion section.
The protocol used for the determination of IL-20 levels has been included in the material and methods section. The phrase “For the determination of the IL-20 levels, we designed a panel, which also included the IL-19 and IL-22 cytokines (Bio-Rad, CA, USA). The levels were detected using Luminex multi-analyte technology (Bio-Plex ProTM , Bio-Rad, CA, USA) according to the manufacturer’s instruction. Before starting the bioassay, the samples were thawed on ice and once ready for use, they were centrifuged at 10000 rpm for 4 minutes. Samples were incubated with antibodies immobilized on color-coded microparticles, washed to remove unbound material, and then incubated with biotinylated antibodies to the molecules of interest. After further washing, the streptavidin-phycoerythrin conjugate that binds to the biotinylated antibodies was added before the final washing step. The Luminex analyzer was used to determine the magnitude of the phycoerythrin-derived signal in a microparticle-specific manner.The data were analyzed using Bio-Plex Manager software. Data were expressed in pg/mL.” has been added in the material and methods section.
Reviewer 2 Report
The work has been improved. I accept the amendments.
I wrote my minor comments with a different font color. If the authors agree with them, they can apply them.

Author Response
Answer:We agree with corrections suggested by the reviewer.
The page: https://www.broadinstitute.org/haploview/haploview, has been added as a reference for Haploview.
The term “in circulation”has been changed to “in circulating blood”
The phrase“As for BMI and VAT/SAT ratio, the individuals with CC genotypes showed a statistically significant negative correlation between these parameters (r2 = 0.021; P = 0.0087) and no such association was found in relation to CT+TTgenotypes (P = 0.579) (Figure 2A). In order to better understand whether adipose tissue levels may impact the IL-20 concentration according to the genotypes of the rs1400986 polymorphism, we performed a correlation analysis in 106 control participants (CC n = 34, CT n = 36 and TT n = 36).” was modified as was suggested by the reviewer.
The terms “ocurring”and “It was found that”have been included.